# Chemical Composition and Functional Properties of Dietary Fibre Concentrates from Winemaking By-Products: Skins, Stems and Lees

**DOI:** 10.3390/foods10071510

**Published:** 2021-06-30

**Authors:** María Ángeles Rivas, Rocío Casquete, María de Guía Córdoba, Santiago Ruíz-Moyano, María José Benito, Francisco Pérez-Nevado, Alberto Martín

**Affiliations:** 1Nutrición y Bromatología, Escuela de Ingenierías Agrarias, Universidad de Extremadura, Avd. Adolfo Suárez s/n, 06007 Badajoz, Spain; mrivasm@unex.es (M.Á.R.); mdeguia@unex.es (M.d.G.C.); srmsh@unex.es (S.R.-M.); mjbenito@unex.es (M.J.B.); fpen@unex.es (F.P.-N.); amartin@unex.es (A.M.); 2Instituto Universitario de Investigación en Recursos Agrarios (INURA), Avd. de la Investigación, Universidad de Extremadura, 06006 Badajoz, Spain

**Keywords:** soluble and insoluble dietary fibre, non-extractable phenolic compounds, antioxidant capacity, neutral sugar, in vitro fermentation

## Abstract

The objective of this study was to evaluate, from a technological and nutritional point of view, the chemical composition and functional properties of the industrial winemaking by-products, namely skins, stems and lees. The chemical and physical characteristics, as well as the functional properties (fat and water retention and swelling capacity, antioxidant capacity, and their prebiotic effect), of the dietary fibre of these by-products were studied. The results showed that the skins, stems, and lees are rich in fibre, with the stem fibre containing the highest amounts of non-extractable polyphenols attached to polysaccharides with high antioxidant activity and prebiotic effect. Lee fibre had the highest water retention capacity and oil retention capacity. The results reveal that winemaking by-products could be used as a source of dietary fibre with functional characteristics for food applications.

## 1. Introduction

Waste from the agri-food industries is a current matter of global concern, with the generation of around 37 million tons of agricultural residues in the world during 2017 [1]. The generation of agricultural waste is particularly of concern in viticulture; it is estimated that 25 kg of waste is produced for every 100 kg of grapes [2]. This involves an economic and ecological problem in the management of the winemaking industries [3,4]. The recovery process plays an important role in the circular economy approach. It improves biomass value where the biorefinery acts as a platform that includes several conversion technologies [5,6].

Industrial winemaking activities produce solid residues such as grape pomace (60% of total wine by-products), which is mainly made up of grape skins (50%), residual pulp and stalks (25%) and seeds (25%) [7]. In addition, another by-product to highlight is the wine lees that are produced during fermentation [8] and account for 25% of the waste produced [9,10,11]. These by-products are rich in dietary fibre (DF), being an important source of soluble polysaccharides [12,13,14,15] as well as antioxidant compounds [16,17,18,19,20]. Therefore, by-products of winemaking can be used to produce ingredients with suitable functional properties for the development of new food products. These by-products were used for novel biscuit formulation as an alternative to DF and phenolic compounds [21] and showed reduced oxidation of seafood and meats [22,23]. In addition, insoluble dietary fibre from grape pomace decreased tannins in red wine by up to 38%, showing its efficacy as a clarifying agent in wines [24].

The functional properties of dietary fibre—including the water retention capacity (WRC), the swelling capacity (Sw), the fat retention capacity (FAC), antioxidant activity and prebiotic activity—are associated with the physicochemical characteristics of cell wall polysaccharides, varying according to their composition [25]. WRC, mainly related to insoluble dietary fibre (IDF), prevents and treats different intestinal disorders by increasing faecal bulk and reducing the gastrointestinal transit time. In food-technological terms, dietary fibre with high WRC can be used as a functional ingredient to avoid syneresis and to modify the viscosity and texture of some formulated foods, whereas dietary fibre with high FAC allows stabilisation of fat in emulsion-based products [26]. Sw in the stomach and an increase in viscosity of the digesta is associated with soluble dietary fibre (SDF), which slows down the absorption of nutrients from the intestinal mucosa and lowers the postprandial blood glucose and insulin responses [27].

The prebiotic effect of dietary fibre is probably the most important functional property. Dietary fibre reaches the colon, where it is fermented by the intestinal microbiota, generating short-chain fatty acids (SCFA), such as butyric, propionic, and acetic acids [28]. These compounds are associated with a wide range of physiological properties, including the improvement of digestive tract disorders [29,30,31] and anticancer activity [32,33].

In this context, the objective of this study was to analyse the chemical composition and the functional properties of skins, stems, and lees as by-products of industrial winemaking and thus offer new opportunities for waste use in the wine industry.

## 2. Materials and Methods

### 2.1. Plant Material

Winemaking by-products used in this work were provided by wineries from the Region of Extremadura, Spain. Winemaking by-products studied included red grape skins, stems, and wine lees from grapes of the Tempranillo variety. Grape skin and stem samples were taken after pressing the grapes. In the case of lees, samples were taken at the end of the fermentation. The samples were freeze-dried (LyoBeta, Telstar, Barcelona, Spain). The parameters of the freeze-drying process were freezed for 4 h at −40 °C and primary drying (8.5 h at −20 °C and 6.5 h at 20 °C) at 400 µbar. The samples were then ground with a grinder and sieved with a fine mesh (max 1 mm). Finally, the samples were vacuum-packed using a vacuum packing machine (Model SAMMIC SV-420, Gipuzkoa, Spain) and stored at room temperature until use. All determinations (Figure 1) were done in triplicate.

### 2.2. Chemical Composition

#### 2.2.1. Moisture and Ash

The moisture and ash determinations were based on methods from AOAC International [34]. Moisture and ash content was determined by drying the samples at 105 and 500 °C, respectively, until a constant weight was achieved.

#### 2.2.2. Crude Protein and Total Fat

Crude protein was determined following the Kjeldahl method [35]. Fat content was determined gravimetrically by extraction with diethyl ether using a Soxhlet apparatus [36].

#### 2.2.3. Determination of Soluble Sugars

Total soluble sugars (TSS) were extracted with distilled water and determined using the sulfuric acid-ultraviolet (UV) method proposed by Albalasmeh et al. [37]. Reducing sugars (RS) were determined by the dinitrosalicylic acid (DNS) method [38]. Calibration was performed with standard solutions of glucose. The results were expressed as g/100 g dry sample.

For the characterisation of soluble sugars, the carbohydrates in 1 mL of solution were converted into alditol acetates and quantified by gas chromatography (Shimadzu 2010 Plus) following the method described by Bastos et al. [39]. A capillary column, DB-225 (30 m × 0.25 mm i.d.; 0.15 µm; Agilent, Santa Clara, CA, USA) and auto-injector (Shimadzu AOC-20i) were used. The temperatures of the FID detector and injector were 240 and 230 °C, respectively. The oven temperature was initially held at 140 °C for 2.5 min, then increased to 200 °C at a rate of 20 °C/min and held for 4.5 min, after which it increased at a rate of 30 °C/min to a final temperature of 220 °C, at which it was held for 18 min. The injection volume was 1 µL with a split ratio 1:10. Helium was used as a carrier gas at a flow rate of 1.49 mL/min. Components were identified by comparing their retention times with those of derivatised standards and quantified using 2-deoxyglucose as an internal standard. The results were expressed as mg/g dry samples.

#### 2.2.4. Determination of Total, Soluble, and Insoluble Dietary Fibre

The total dietary fibre (TDF) of winemaking by-products was measured following the standard enzymatic-gravimetric method [40]. First, 1 g of dry sample was mixed into 50 mL of distilled water. Then, samples were digested with 200 µL of α-amylase (Sigma-Aldrich, St. Louis, MO, USA) at 80 °C for 1 h with constant agitation. After digestion, 100 µL of amyloglucosidase solution (50 mg/mL) (Sigma-Aldrich) was added and the mixture was kept at 60 °C for 3 h. Next, the pH was adjusted to 7.0 with NaOH 10% (*w*/*v*), followed by incubation with 200 µL of protease (Sigma-Aldrich) at 80 °C for 1 h. Finally, digested samples were vacuum filtered with cellulose-free filters (Whatman glass microfiber filters, 934-AHTM). The solid fraction contained in the filter represented insoluble dietary fibre (IDF). To precipitate the soluble dietary fibre (SDF), 4 volumes of 96% ethanol were added to the filtrate at 60 °C. Both fractions of fibre were dried overnight at 45 °C in an oven and were then weighed. The TDF was calculated as the sum of IDF and SDF. The results were expressed as g/100 g dry sample. All experiments were conducted in triplicate.

### 2.3. Characterisation of Insoluble Fibre: Cellulose, Hemicellulose, and Lignin

The determination of cellulose, hemicellulose and lignin was carried out by calculating the fractions of neutral detergent fibre (NDF) and acid detergent fibre (ADF) [41] in a fibre extractor (Dosi-Fiber, J.P. Selecta, Abrera, Spain). The ADF determination consisted of boiling the sample with cetyltrimethylammonium bromide in an acid medium with subsequent filtration and washing of the residue. This method resulted in a good estimate for cellulose and lignin. For NDF determination, the sample was treated with a hot solution of sodium lauryl sulphate with a subsequent gravimetric determination of the residue. This method gives a good estimate of insoluble fibre (cellulose, hemicellulose, and lignin). The difference between NDF and ADF was the hemicellulose content. The cellulose, hemicellulose, and lignin contents were expressed as mg/100 g of fibre.

### 2.4. Characterisation of Soluble Fibre: Neutral Sugar and Pectins

Extraction of dietary fibre from the winemaking by-products was carried out using a modification of the double residue method in alcohol to determine the alcohol-insoluble residue (AIR) described by Femenia et al. [42]. Briefly, 3 replicates (5 g each) of the dry sample were homogenised with 85% (*v*/*v*) ethanol. The mixture was boiled on a shaker for 10 min, and then the solid residue was collected using a Büchner funnel with cellulose-free filters (Whatman, 934-AHTM glass microfiber filters). This process was repeated twice, the last time with absolute ethanol. Finally, the insoluble solid residue was washed with acetone, and the excess solvent was removed after 24 h at room temperature.

Total neutral sugars (TNS) were released through a process of hydrolysis of the fibre using 12 M sulfuric acid (3 h at room temperature and 100 °C for 1 h) and were determined, as D-glucose equivalent, with the anthrone method proposed by Van Handel [43].

For the characterisation of neutral sugars (rhamnose, fucose, arabinose, xylose, mannose, galactose, and glucose), neutral sugars were converted into their alditol acetates and quantified by gas chromatography as described in Section 2.2.2. The content of each individual neutral sugar was expressed as mg/g.

The uronic acid content was determined spectrophotometrically by the m-hydroxydiphenyl method [44] with galacturonic acid as standard and 3,5-dimethylphenol as reagent. The results were expressed as mg galacturonic acid/g AIR.

### 2.5. Functional Properties of the Fibre

#### 2.5.1. Swelling, Water Retention Capacity and Fat Retention Capacity

Sw, WRC and FAC determinations were carried out following the method described by Garau et al. [45]. For Sw, 0.1 g of AIR was mixed with 10 mL of distilled water on a calibrated cylinder. After 24 h incubation at room temperature (20–25 °C), the increase in volume was measured, and the results were expressed as mL water/g AIR. For WRC, 0.2 g of AIR was hydrated in 10 mL of distilled water and left for 24 h at room temperature (20–25 °C). The next day, the sample was centrifuged at 2000× *g* for 25 min. After centrifugation, the supernatant was decanted, and the resulting solid residue was weighed. WRC was expressed as g water/g AIR. For FAC, the process was the same as for WRC determination, using 5 mL of sunflower oil in place of the 10 mL of water. The results were expressed as g oil/g AIR.

#### 2.5.2. Non-Extractable Polyphenols and Antioxidant Activity

The determination of the antioxidant capacity of non-extractable polyphenols bound to AIR was achieved according to the method described by Arranz et al. [46] with some modifications. A total of 0.5 g of AIR was mixed with 20 mL of a methanol/water solution (50: 50) acidified with hydrochloric acid to pH 2. The mixture was incubated with stirring for 1 h and subsequently centrifuged for 20 min at 2500× *g*. This process was repeated with an acetone/water solution (70: 30). The excess solution was removed by heating at 37 °C in a rotary evaporator under vacuum (Hei-VAP Precision, Heidolph, Germany). The resultant residue was resuspended in 30 mL of distilled water. The total non-extractable polyphenols bound to AIR were determined using the Folin-Ciocalteure agent [47] in a UV-1800 spectrophotometer (Shimadzu Scientific Instruments, Columbia, MD, USA). Gallic acid was used as the standard. The results were expressed as mg gallic acid equivalent (GAE)/100 g AIR.

The antioxidant capacity of the samples was evaluated by 2 antioxidant assay methods: the 2,2-diphenyl-1-picrilhydrazyl (DPPH) depletion method according to the procedure of Teixeira et al. [48], and the capacity to remove the 2,2′-azinobis (3-ethylbenzothiazoline-6-sulfonic acid) (ABTS) radical according to the method of Re et al. [49]. Trolox was used as the standard, and the results were expressed as mg Trolox equivalent/100 g AIR.

#### 2.5.3. In Vitro Prebiotic Capacity

The prebiotic capacity of the soluble dietary fibre extracts on lactic acid bacteria (LAB) associated with fermented food products was determined using the method described previously by Ruiz-Moyano et al. [50]. For this study *Lactococcus lactis* (CECT 188), *Lactobacillus curvatus* (CECT 904), *Lactobacillus sakei* (CECT 5765 and CECT 980), *Lactobacillus brevis* (CECT 815), *Lactobacillus plantarum* (G1LB5; [46]), *Lactobacillus casei* (HL 245 and HL 233; [51]) and *Enterococcus faecium* (SE 906 and SE 920; [52]) were used. Prior to the assay, LAB were grown in de Man–Rogosa–Sharpe broth (MRS; Condalab, Madrid, Spain) at 37 °C for 24 h. The strains were tested for growth in the presence of solubilised AIR (autoclaved at 121 °C for 16 min followed by an ultrasound treatment for 1 h). The absence of free sugars in each solubilised AIR was previously checked. For each bacterial strain, 5 μL of suspension was inoculated in 200 μL of semi-solid MRS medium containing 0.125 g/L agar without glucose and supplemented with 2 g/L of each sterile filtered dietary fibre extract as the sole carbohydrate source. The positive control for growth consisted of semi-solid MRS supplemented with 2 g/L glucose, whereas the negative control was a carbohydrate-free semi-solid MRS. Turbidity was measured in a fluorescence microplate reader (FLUOstar OPTIMA F, BMG Labtech, Ortenberg, Germany), growth was carried out for 48 h at a temperature of 37 °C, with readings at a wavelength of 570 nm every 1 h. For each strain, the ability to grow was evaluated by comparing the percentage growth in each extract with the positive control.

### 2.6. Statistical Analysis

Statistical analysis of the data was carried out using SPSS for Windows, version 21.0 (IBM Corp., Armonk, NY, USA). Descriptive statistics of the data were determined, and the differences within and between groups were studied by one-way analysis of variance (ANOVA) and separated by Tukey’s honest significant difference test (*p* ≤ 0.05). Principal component analysis (PCA) was performed on the correlation matrix of the variables.

## 3. Results and Discussion

### 3.1. General Chemical Composition

The results of the chemical composition of winemaking by-products (skins, stems and lees) are shown in Table 1. The stem samples showed higher moisture values (*p* ≤ 0.05) than the skin and the lee samples, reaching 16.26%. The lowest moisture value corresponded to the skin samples, with a value of 7.66%. However, the ash content of the lee samples was 13.18%, which was significantly higher than the amount found in other samples, ranging from 6.08% for stems up to 8.12% for the skin. The samples with the highest crude protein content were lees (22.32%), followed by skin and stem samples, with values of 12.24% and 7.94%, respectively. The fat content ranged from 1.95 to 4.85% (Table 1), not showing significant differences between samples (skins, stems and lees). The fact that the lees were the by-product with the highest percentage of proteins was explained by the nature of the residue; lees were mainly made up of the remains of the yeast autolysis during the process of fermentation. In the same way, Rubio et al. [53] also found higher protein content in lees, followed by skin and stems. The values obtained from skin samples agreed with those published by Deng et al. [54]. However, the stem and lees values differed from those found by González-Centeno et al. [55] and Bordiga [56]. The composition depended on numerous parameters related to the types of grapes and in the case of the lees the type of yeast used, and to the vinification method, resulting in a wide compositional heterogeneity, as shown by data reported in the literature [57].

#### 3.1.1. Total Soluble Sugars

Table 1 also shows the TSS content of the winemaking by-products. The values of total and reducing sugars (consisting of similar amounts of glucose and fructose) were highest in stem samples, with 22.01 and 19.60 g/100 g, respectively, because this by-product does not undergo fermentation [53]. On the contrary, the lees extracted when the malolactic fermentation had finished contained the lowest TSS content because they were transformed into alcohol.

#### 3.1.2. Total, Insoluble and Soluble Dietary Fibre

The samples of skin, stem and lees studied were rich in TDF (71.39–82.32 g/100 g) and in IDF (Table 1). IDF has been described as the predominant dietary fibre fraction for winemaking by-products [58]. The IDF content from samples ranged from 67.68 to 78.43 g/100 g, showing higher values for lees and skin (Table 1). The content of SDF was similar for each of the by-products studied, with values between 3.71 and 4.13 g/100 g. The values obtained from TDF agreed with those published by other authors for skin and stem samples [59,60]. However, the composition of dietary fibre in winemaking lees continued to create controversy: some authors affirm that lees do not contain a significant fraction of dietary fibre [61], while other authors state that the content is around 22–50% [53]. The high dietary fibre content found in this work for lees has not been reported in previous publications. The high variability of these data was probably due to the different recovery methods, winemaking procedures applied, and varieties of grape.

### 3.2. Fibre Constituents: Cellulose, Hemicellulose, Lignin, Neutral Sugars, and Pectins

In skin, stems and lee samples, insoluble dietary fibre is mainly lignin (47.31, 29.83, and 44.41 g/100 g, respectively), followed by hemicellulose and cellulose (Table 2). The values obtained for lignin, cellulose and hemicellulose in the skin, stems and lee samples were similar to those reported in the literature [53,62,63,64].

The most abundant sugars in the AIR of the skin and the stems (Table 2) were uronic acids (which indicate the pectin content), followed by glucose and fucose in the case of the skin, and mannose in the stems, while arabinose, xylose, galactose and rhamnose were the minority sugars. In the case of lees, glucose was the major monosaccharide, followed by rhamnose and mannose, with concentrations around 78, 50 and 35 g/100 g, respectively. Uronic acids, fucose and arabinose values were lower in the lees than in skin and stems (Table 2). The profile found for the sugar composition of skin and stem agreed in general with the results obtained by other authors [55,65], although other authors have described pectin (measured as galacturonic acid equivalent) and glucose as the major skin and stem cell wall components [66]. High glucose levels could be related to high glucan and xyloglucan values [66,67,68].

With respect to xylose and fucose, higher amounts were found in skin samples than those described by other authors for these winemaking by-products [67,69,70]. The presence of xylose and fucose could indicate a higher amount of hemicellulosic polysaccharides in grape skins [71]. Ortega-Regules et al. [72] showed that the neutral sugar profile of the vinification by-products was highly variable, depending on the cultivar used. The degree of maturation of the grapes, their geographical origin and oenological techniques were also responsible for the differences found in the chemical composition of the by-products [66,73].

On the other hand, there were few studies on the profile of neutral sugars in winemaking lees, although some authors have reported that the main monosaccharides of lees were glucose, mannose and rhamnose [74,75], which agreed with the data of our study.

### 3.3. Functional Properties of the Fibre

#### 3.3.1. Swelling, Water Retention Capacity and Oil Retention Capacity

The functional properties of dietary fibres (Sw, WRC and FAC) were important in determining suitability for application as functional ingredients in foods. The results of these functional property analyses of winemaking by-products are shown in Table 3.

The Sw values, related to the porosity of the fibre [76], were significantly higher for the lee samples than for skin samples. However, the values of WRC and FAC did not show differences between the three by-products studied (Table 3). These values were similar to those published by other authors for winemaking by-products [55]. However, the stem samples showed lower values of WRC than those reported in previous studies, ranging from 5.5 to 10.7 g water/g AIR [55]. These differences may be due to the nature of the plant material used, in particular the IDF content. The structure and chemical composition of the fibre were known to play an important role in the kinetics of water absorption [77,78]. The high FAC values obtained in the stem and lee samples are beneficial because they are associated with oil retention during the digestion of food and reduce serum cholesterol levels [79].

In summary, the lee samples exhibited the best characteristics for use as a functional food ingredient, although these properties not only depend on the type of soluble fibre, but also on the particle size distribution and the characteristics of the surface [80]. In addition, the particle size may cause changes in the functional properties of dietary fibre, thus smaller particle sizes showed higher values of WRC and Sw [80].

#### 3.3.2. Non-Extractable Polyphenols and Antioxidant Activity

The phenolic compound content in AIR (N-EPC) and the antioxidant capacity, determined by two methods (DPPH and ABTS), are also shown in Table 3. Among the three AIRs, stem AIR presented the highest N-EPC value of 138.63 mg GAE/100 g, whereas lees AIR had the lowest N-EPC value of 44.64 mg GAE /100 g. This difference may be due to the lignin content of each of the samples (Table 2). Bender et al. [62] observed that the concentration of phenolic compounds increased with reduced lignin content because of the breakdown of the lignin structure. Changes in the structure of the fibre affected the non-extractable phenolic compound content by loosening the hydrogen bonds and causing delignification [81].

Regarding DPPH and ABTS values, significant differences were found between the samples of by-products studied (Table 3). The stem samples showed the highest antioxidant activity both by the DPPH method (6093.01 mg Trolox/100 g) and ABTS (5682.74 mg Trolox/100 g), followed by skin and lee samples in agreement with the total phenolic content observed in samples (Table 3). In fact, there was a highly significant correlation (r = 0.961; *p* < 0.01) between the content of phenolic compounds of the fibre and its antioxidant activity by the DPPH method. The results reflect that the increase in antioxidant activity was promoted by the non-extractable total phenolic content. This mechanism has been corroborated by previous studies [82,83,84]. Ferri et al. [82] reported that seeds and raw skin extracts of several grape cultivars exhibited a high antioxidant capacity measured by the ABTS method, which was positively correlated with the total polyphenol content.

#### 3.3.3. In Vitro Prebiotic Capacity

Table 4 shows the capacity of the 10 LAB strains tested for in vitro growth on three extracts of dietary fibre, observing differences in the growth of the strains according to the fibre used. Even though we expected higher bacterial growth on culture media with the skin fibre, due to the presence of fermentable fibre components such as pectins and hemicelluloses, we only observed low growth (Table 4). This growth was similar to growth on culture media with the lees fibre. The culture media enriched with the dietary fibre extracted from the stem samples presented the highest percentage of growth for all strains studied. The superior growth of bacteria with stem fibre could be related to the presence of the non-extractable polyphenols (Table 3). The data obtained could suggest that the studied strains use some phenols for their metabolism. This finding agrees with the study by Landete et al. [85]. The authors demonstrated that bacteria such as *Lactobacillus* spp. can degrade phenolic compounds to other molecules with high added value. Nonetheless, more studies would be needed in relation to the structures of the oligosaccharides present to know the potential of the grape stem as prebiotics. *E. faecium* SE 906E had the highest growth percentage (76.79%), followed by *L. sakei* CECT 5765 (67.11%) and *E. faecium* SE 920 (62.46%) with moderate growth, showing differences from the rest of the strains (*p* ≤ 0.05).

### 3.4. Multivariate Analysis of the Parameters Related to Dietary Fibre Extracted from Skin, Stems and Lees

PCA was carried out for the entire set of dietary fibre data to obtain an interpretable overview of the main information. Figure 2 shows the two-way loadings and score plots, where PC2 was plotted against PC1, explaining more than 90% of the total variance. Higher values for AIR, N-EPC, antioxidant activity (DPPH and ABTS) and bacterial growth were clearly correlated and explained the positive axis of PC1, which was related to the stem samples. Therefore, the principal component analysis confirmed the results shown previously, indicating that the increase in antioxidant activity was promoted by the N-EPC content and that the culture media enriched with the dietary fibre extracted from the stem samples showed the highest percentage of growth. On the contrary, TNS and functional properties (WRC and FAC) explained the negative axis of PC1 and were associated with lee samples. The second PC was mainly explained by neutral sugars (galactose, arabinose, and xylose) located in the extreme of the negative axis, relating to high values in skin samples.

## 4. Conclusions

This study presents a complete physical and chemical characterisation of three winemaking by-products: skins, stems and lees. These by-products, rich in fibre, were demonstrated to have useful functional properties and a high non-extractable concentration of phenolic compounds. The stems have a high concentration of non-extractable polyphenols attached to polysaccharides, high antioxidant activity, and are a good substrate for bacterial growth. However, lees showed a high concentration of total neutral sugar and fibre with a good aptitude for the functional properties WRC and FAC. The skins gave high concentrations of galactose, arabinose and xylose. The results reveal that winemaking by-products can be considered sources of high-quality dietary fibre for food applications with good functional characteristics. However, further studies on the structures of the oligosaccharides involved would be necessary for a better understanding of grape stems as prebiotics.

## Figures and Tables

**Figure 1 foods-10-01510-f001:**
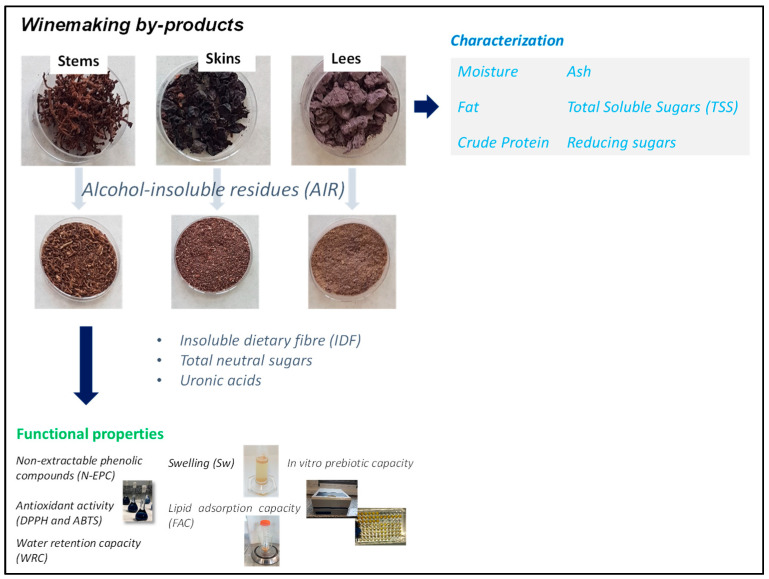
Graphic diagram of the study design.

**Figure 2 foods-10-01510-f002:**
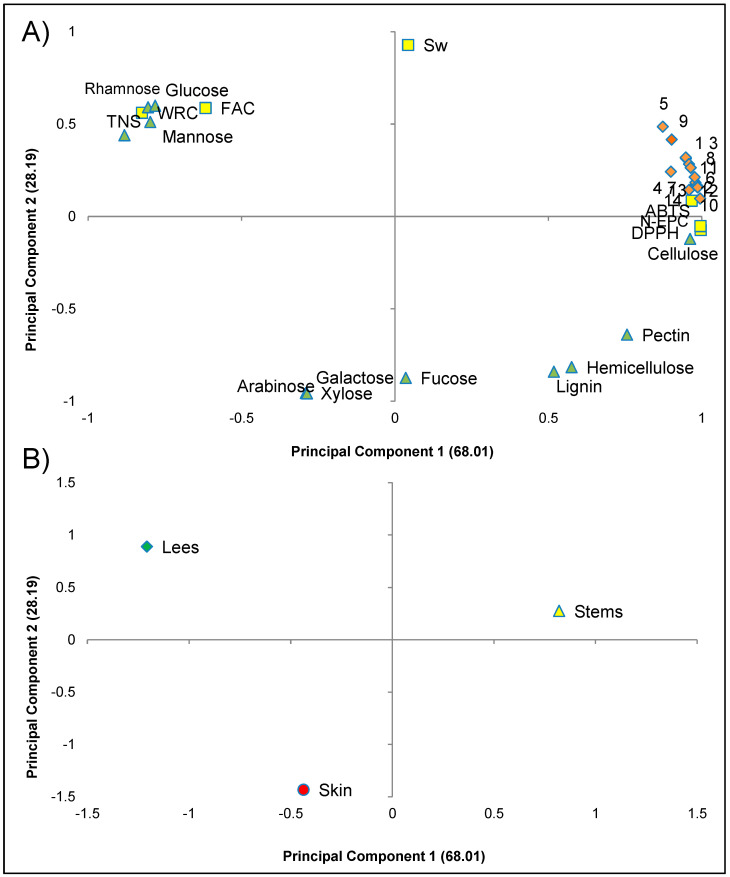
Principal component analysis of the analytical results of dietary fibre from skin, stems and lee samples. Loading plot (**A**). TNS, total neutral sugar; N-EPC, non-extractable phenolic compounds; DPPH and ABTS, antioxidant activity; Sw, swelling; WRC, water retention capacity; FAC, fat retention capacity; Bacterial growth: 1, *Lactobacillus curvatus* CECT 904; 2, *Lactococcus lactis* CECT 188; 3, *Lactobacillus sakei* CECT 5765; 4, *Lactobacillus sakei* CECT 980; 5, *Lactobacillus brevis* CECT 815; 6, *Lactobacillus plantarum* G1LB5; 7, *Lactobacillus casei* HL 245; 8, *L. casei* HL 233; 9, *Enterococcus faecium* SB 906; 10, *E. faecium* SB 920. Score plot (**B**). Skin, stem, and lees sample.

**Table 1 foods-10-01510-t001:** Chemical composition of skins, stems, and lees (g/100 g dry weight).

Parameters	Skins	Stems	Lees
	Mean SD ^1^	Mean SD	Mean SD
Moisture	7.66 ± 0.15 ^a^	16.26 ± 0.16 ^c^	9.09 ± 0.27 ^b^
Ash	8.12 ± 1.40 ^a^	6.08 ± 1.18 ^a^	13.18 ± 0.92 ^b^
Protein	12.24 ± 0.88 ^b^	7.94 ± 0.35 ^a^	20.32 ± 0.75 ^c^
Fat	4.24 ± 0.93	1.95 ± 0.39	4.85 ± 1.15
TSS	3.65 ± 0.35 ^a^	22.01 ± 1.56 ^b^	1.63 ± 0.09 ^a^
Reducing sugars	1.78 ± 0.08 ^a^	19.60 ± 2.37 ^b^	0.49 ± 0.05 ^a^
Glucose	0.90 ± 0.73 ^a^	10.62 ± 11.22 ^b^	<0.5 *^a^
Fructose	0.79 ± 0.54 ^a^	11.71 ± 12.01 ^b^	<0.5 *^a^
TDF	82.30 ± 2.71 ^b^	71.39 ± 1.01 ^a^	82.32 ± 1.69 ^b^
IDF	78.18 ± 2.91 ^b^	67.68 ± 0.95 ^a^	78.43 ± 1.32 ^b^
SDF	4.13 ± 0.20	3.71 ± 0.06	3.90 ± 0.37

TSS, total soluble sugars; TDF, total dietary fibre; IDF, insoluble dietary fibre; SDF, soluble dietary fibre. ^1^ SD, standard deviation. * The limit of detection was 0.5 mg/g. ^a,b,c^ Values with different superscripts are significantly different (*p* ≤ 0.05) between samples.

**Table 2 foods-10-01510-t002:** Fibre constituents of skins, stems and lees.

Parameters	Skins	Stems	Lees
	Mean SD ^1^	Mean SD	Mean SD
IDF (g/100 g)			
Hemicellulose	22.57 ± 2.16	17.18 ± 0.22	28.79 ± 9.25
Cellulose	7.24 ± 0.04 ^a^	14.55 ± 3.59 ^a^	30.03 ± 0.54 ^b^
Lignin	47.31 ± 3.61 ^b^	29.83 ± 3.27 ^a^	44.41 ± 5.91 ^b^
Total neutral sugars of AIR (mg/g)	114.71 ± 19.04 ^b^	84.82 ± 16.76 ^a^	156.04 ± 8.82 ^c^
Rhamnose	<0.5 *^a^	<0.5 *^a^	50.33 ± 0.95 ^b^
Fucose	32.37 ± 2.51 ^b^	12.40 ± 11.31 ^ab^	2.32 ± 0.47 ^a^
Xylose	11.76 ± 1.15 ^b^	<0.5 *^a^	<0.5 *^a^
Mannose	19.61 ± 0.77 ^a^	17.94 ± 4.42 ^a^	35.63 ± 1.32 ^b^
Glucose	26.07 ± 1.06 ^a^	28.07 ± 5.30 ^a^	78.30 ± 1.45 ^b^
Galactose	7.56 ± 1.97 ^b^	<0.5 *^a^	<0.5 *^a^
Arabinose	17.90 ± 0.66 ^b^	4.08 ± 0.57 ^a^	5.30 ± 1.76 ^a^
Uronic acids of AIR (mg/g)			
Uronic acids (pectin)	31.37 ± 1.47 ^b^	31.09 ± 3.17 ^b^	12.37 ± 0.53 ^a^

IDF, insoluble dietary fibre; AIR, alcohol-insoluble residue. ^1^ SD, standard deviation. * The limit of detection was 0.5 mg/g. ^a,b,c^ Values with different superscripts are significantly different (*p* ≤ 0.05) between samples.

**Table 3 foods-10-01510-t003:** Functional properties and non-extractable phenolic compounds (N-EPC) of dietary fibre from skins, stems and lees.

Parameters	Skins	Stems	Lees
	Mean SD ^1^	Mean SD	Mean SD
Sw (mL water/g)	6.55 ± 0.03 ^a^	7.76 ± 0.54 ^ab^	8.43 ± 0.47 ^b^
WRC (g water/g)	4.57 ± 0.21	4.61 ± 0.34	9.11 ± 2.39
FAC (g oil/g)	3.76 ± 0.27	5.48 ± 0.51	5.48 ± 0.88
N-EPC (mg GAE/100 g)	92.83 ± 15.76 ^b^	138.63 ± 28.68 ^c^	44.64 ± 4.35 ^a^
DPPH (mg Trolox/100 g)	3823.68 ± 63.65 ^b^	6093.01 ± 376.40 ^c^	2049.23 ± 33.73 ^a^
ABTS (mg Trolox/100 g)	4205.82 ± 307.11 ^b^	5682.74 ± 308.97 ^c^	2395.50 ± 671.69 ^a^

Sw, swelling water capacity; WRC, water retention capacity; FAC, oil retention capacity; GAE, gallic acid equivalents; DPPH and ABTS, antioxidant capacity. ^1^ SD, standard deviation. ^a,b,c^ Values with different superscripts are significantly different (*p* ≤ 0.05) between samples.

**Table 4 foods-10-01510-t004:** Percentage of growth of lactic acid bacteria (LAB) strains, with respect to the positive control, on soluble dietary fibre extracts of skins, stems and lees.

Microorganisms	Skins	Stems	Lees
	Mean SD *	Mean SD	Mean SD
*Lactobacillus curvatus* CECT 904	11.04 ± 0.86 ^a1,2^	25.83 ± 0.77 ^b1^	11.04 ± 0.86 ^a1,2^
*Lactococcus lactis* CECT 188	9.28 ± 0.76 ^b1^	26.74 ± 0.06 ^c1^	6.00 ± 0.34 ^a1^
*Lactobacillus sakei* CECT 5765	28.74 ± 2.59 ^a4,5^	67.11 ± 0.85 ^b3,4^	27.11 ± 0.05 ^a4^
*L. sakei* CECT 980	20.08 ± 1.08 ^b2,3,4^	44.61 ± 1.58 ^c2^	14.08 ± 0.57 ^a2,3^
*Lactobacillus brevis* CECT 815	9.42 ± 0.64 ^a1^	29.90 ± 0.58 ^c1^	14.12 ± 0.43 ^b2,3^
*Lactobacillus plantarum* G1LB5	10.58 ± 0.89 ^a1,2^	33.50 ± 8.62 ^b1,2^	8.17 ± 0.54 ^a1,2^
*Lactobacillus casei* HL 245	10.30 ± 0.42 ^a1,2^	39.65 ± 3.51 ^b1,2^	12.19 ± 4.08 ^a1,2,3^
*L. casei* HL 233	15.52 ± 6.92 ^a1,2,3^	47.39 ± 0.08 ^b2^	13.28 ± 4.65 ^a1,2,3^
*Enterococcus faecium* SE 906	30.46 ± 4.58 ^a5^	76.79 ± 0.48 ^b4^	23.02 ± 3.79 ^a4^
*E. faecium* SE 920	22.49 ± 0.68 ^b3,4,5^	62.46 ± 3.11 ^c3^	12.11 ± 0.17 ^a1,2,3^

* SD, standard deviation. ^a,b,c^ Values with different superscripts are significantly different (*p* ≤ 0.05) between samples. ^1,2,3,4,5^ Values with different subscripts are significantly different (*p* ≤ 0.05) between strains in one sample. Negative growth (<20%). Slight growth (20–40%), Moderate growth (40–70%). High growth (>70%).

## Data Availability

Not applicable.

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
