# Peer review of "Chemical Composition and Functional Properties of Dietary Fibre Concentrates from Winemaking By-Products: Skins, Stems and Lees"

_foods, 2021, doi:10.3390/foods10071510_

Round 1
Reviewer 1 Report
The character of this paper is novel in the perspective of circular economy and the subject is well described and discussed. In Introduction the authors should add a desriptive part on winemaking by-products and related properties and profile of bioactive compounds, in this regar related references should be added in the context of biorefinery approach such as:
Lucarini et al. Grape Seeds: Chromatographic Profile of Fatty Acids and Phenolic Compounds and Qualitative Analysis by FTIR-ATR Spectroscopy. Foods. 2019, 9(1). pii: E10. doi: 10.3390/foods9010010
Chowdhary P, Gupta A, Gnansounou E, Pandey A, Chaturvedi P. Current trends and possibilities for exploitation of Grape pomace as a potential source for value addition. Environ Pollut. 2021 Jun 1;278:116796. doi: 10.1016/j.envpol.2021.116796.
Details on samples and a graphical scheme of study design should be added.
Figure 1 should be better described in the text
Conclusion and Future remarks should be implemented
Author Response
Response to Reviewer 1 Comments
Reviewers' comments:
We thank the reviewers and editor for their dedication and comments.
The character of this paper is novel in the perspective of circular economy and the subject is well described and discussed. In Introduction the authors should add a descriptive part on winemaking by-products and related properties and profile of bioactive compounds, in this regard related references should be added in the context of biorefinery approach such as:
Lucarini et al. Grape Seeds: Chromatographic Profile of Fatty Acids and Phenolic Compounds and Qualitative Analysis by FTIR-ATR Spectroscopy. Foods. 2019, 9(1). pii: E10. doi: 10.3390/foods9010010
Chowdhary P, Gupta A, Gnansounou E, Pandey A, Chaturvedi P. Current trends and possibilities for exploitation of Grape pomace as a potential source for value addition. Environ Pollut. 2021 Jun 1;278:116796. doi: 10.1016/j.envpol.2021.116796.
We appreciate the reviewer’s contribution. The introduction has been modified as suggested by the reviewer.
Details on samples and a graphical scheme of study design should be added.
Details on samples and a graphical scheme of study design has been added.
Figure 1 should be better described in the text
According to reviewer comment, the text on the figure of principal component analysis has been expanded.
Conclusion and Future remarks should be implemented.
Conclusion and Future remarks have been implemented
Reviewer 2 Report
Winemaking produces a large quantity of waste and by-products. In recent years, winery by-products have been considered as a promising alternative source of bioactive compounds (dietary fibers, phenols, proteins, lipids, hydrocolloids).
Why were the samples dried prior to the determinations? There are studies that show that in the case of dried products it is easier to extract bioactive ingredients (than from fresh raw materials) and more of them go into solution. Hence the results can be overstated. Moreover, freeze-drying is an expensive technique. And then what about the commercialization of the product?
Line 2-3. The title does not match aim of the work. Please think about it.
Line 38-40. New food products? Expand. Please list the individual applications in food production.
Line 68-69. No relevant data. Please provide the parameters of freeze-drying as well as grinding (including equipment used). Were the ground samples additionally sieving? Has the particle size distribution been investigated?
Line 76-77. Provide literature for fat content determination.
Line 89 and anywhere. Unit "minute" replace "min".
Line 100 and anywhere. Unit "hour" replace "h".
Line 171. "according to the method of [45]". Enter the author's name.
Line 213-214. Other authors? The authors only provided a reference to one publication. Report the values obtained by Deng et al. [50].
Line 297-298. Give the values obtained by other authors.
Line 304-306. How does particle size and structure affect the functional properties of the fiber? What is the influence of the drying method on the subsequent grinding of the product? What was the degree of grinding of the material?
Line 329-336. Dietary fiber, as well as polyphenols, can also exert antimicrobial action. Which could result in lower bacterial growth.
Reference style: The literature cited in the text is most often multi-author (more than 2 authors), therefore, in addition to the name of the first author, the note et al. Example:. "Albalasmeh et al. [33]" not "Albalasmeh [33]". If there are only two authors you should use “Arnous and Meyer [63]“ Make corrections throughout the text.
Editing mistakes: 24-hourincubationat, of1.49 mL/minute, ahigh
Author Response
Winemaking produces a large quantity of waste and by-products. In recent years, winery by-products have been considered as a promising alternative source of bioactive compounds (dietary fibers, phenols, proteins, lipids, hydrocolloids).
Why were the samples dried prior to the determinations? There are studies that show that in the case of dried products it is easier to extract bioactive ingredients (than from fresh raw materials) and more of them go into solution. Hence the results can be overstated. Moreover, freeze-drying is an expensive technique. And then what about the commercialization of the product?
We agree with the reviewer that the freeze-drying is a common step in the extraction protocols of bioactive compounds in food products. It guarantees the standardization and comparison of the results from different subtracts such as the by-products studied in this work. The directly extraction from fresh products involve a great variability associated to the moisture and the partial loss of the more hydrophobic compounds. So, under the analytical viewpoint, the we consider that the freeze-drying of the by-products is adequate for the objectives of the work. For de commercial viewpoint, the freeze-drying of the byproducts supposes a relevant advantage for the standardization of the functional extracts despite to increase of price.
Line 2-3. The title does not match aim of the work. Please think about it.
The title has been modified.
Line 38-40. New food products? Expand. Please list the individual applications in food production.
The applications in food production have been included.
Line 68-69. No relevant data. Please provide the parameters of freeze-drying as well as grinding (including equipment used). Were the ground samples additionally sieving? Has the particle size distribution been investigated?
We thank the reviewer´s comment. The sentence has been modified including the data suggested by the reviewer.
Line 76-77. Provide literature for fat content determination.
The reference for fat content determination has been added.
Line 89 and anywhere. Unit "minute" replace "min".
"minute" has been replaced "min".
Line 100 and anywhere. Unit "hour" replace "h".
"hour" has been replaced "h".
Line 171. "according to the method of [45]". Enter the author's name.
The author's name has been included.
Line 213-214. Other authors? The authors only provided a reference to one publication. Report the values obtained by Deng et al. [50].
We thank the reviewer´s comment. The sentence has been modified.
Line 297-298. Give the values obtained by other authors.
The values obtained by other authors has been included.
Line 304-306. How does particle size and structure affect the functional properties of the fiber? What is the influence of the drying method on the subsequent grinding of the product? What was the degree of grinding of the material?
The particles size can cause corresponding changes in the functional properties of dietary fiber, smaller particle sizes showed higher values of WRC and Sw. The samples were ground using a mincer and screened using a fine mesh (max 1 mm). These clarifications have been included in the manuscript.
Line 329-336. Dietary fiber, as well as polyphenols, can also exert antimicrobial action. Which could result in lower bacterial growth.
We agree with the reviewer´s about those polyphenols can exert antimicrobial action. However, according to the results obtained the non-extractable total phenolic content was correlated bacterial growth. The data obtained could suggest that the studied strains use some phenols for their metabolism. This finding agrees with the study by Landete et al. 2007, the authors demonstrated that bacteria such as Lactobacillus spp. can degrade phenolic compounds to other molecules with high added value.
Reference style: The literature cited in the text is most often multi-author (more than 2 authors), therefore, in addition to the name of the first author, the note et al. Example:. "Albalasmeh et al. [33]" not "Albalasmeh [33]". If there are only two authors you should use “Arnous and Meyer [63]“ Make corrections throughout the text.
We thank the reviewer´s comment. The reference style has been modified.
Editing mistakes: 24-hourincubationat, of1.49 mL/minute, ahigh
The mistakes have been corrected.
Reviewer 3 Report
In this paper a complete physical and chemical characterisation of three winemaking by-products: skins, stems, and lees from grapes of the Tempranillo variety, Spain were investigated.
In my opinion, the paper can be accepted for publication after minor revision:
- DPPH and ABTs give the same information, there are based on the different synthetic radial but they give same information, free radical scavenging activity. For this reason, they are not analysed together but in combination with other antioxidant assays (FRAP or ORAC).
- Some parts of the paper (especially Conclusion) should be focused on the obtain results. Claiming that analysed materials have “…a high concentration of phenolic compounds“ is incorrect, since these were not analysed in the paper. Only total non-extractable phenolic
Author Response
In this paper a complete physical and chemical characterisation of three winemaking by-products: skins, stems, and lees from grapes of the Tempranillo variety, Spain were investigated.
In my opinion, the paper can be accepted for publication after minor revision:
DPPH and ABTs give the same information, there are based on the different synthetic radial but they give same information, free radical scavenging activity. For this reason, they are not analysed together but in combination with other antioxidant assays (FRAP or ORAC).
We agree with the reviewer that the used of several antioxidant assays implement the information about the mechanism associated to this activity in the by-products. In the case of DPPH and ABTs assays, both provide information about free scavenging activity but the affinity of this radicals depends of the hydrophobic properties of the compounds checked as it is patent in the differences of the results obtained with these assays. So, DPPH and ABTs assays also provide complementary information about antioxidant activity of the extracts.
Some parts of the paper (especially Conclusion) should be focused on the obtain results. Claiming that analysed materials have “…a high concentration of phenolic compounds“ is incorrect, since these were not analysed in the paper. Only total non-extractable phenolic
We appreciate the reviewer’s contribution. The manuscript has been modified including the comments suggested by the reviewer.
Reviewer 4 Report
The manuscript shows interesting and novel results related to the chemical composition and especially physical and functional characteristics of dietary fibres form winemaking by-products (skins, stems, and lees). The study was well designated with the uses of reliable analytical methods. The results are clearly presented with relatively sufficient discussion and explanation.
There are some minor points should be considered/clarified:
- The number of the replicates of the experiments should be mentioned in the Methods
- Table 1: the sum of components calculated based on “g/100 g dry weight” was higher than 100g. This is not reasonable so double-check or explanation for this result must be done. As mentioned in lines 240-243, the dietary fibre content in this study is surprisingly higher than that reported in previous studies with the similar materials. Is there any problem with operation of experiments or calculation of related results?
- Table 2: The sum of IDF components is more than 100 g. Thus double-check or explanation for this result must be done.
- Table 3: Double-check should be done for the statistical comparison of WRC among the samples
- Section 3.3.2: A comparison of phenolic content and antioxidant capacity of materials used in this study with those of some well-known phenolic-rich and strong antioxidant sources would help reinforce the conclusions of the phenolic levels and antioxidant strength of the materials.
- Figure 1: Symbol A is hidden, correct it pls.
- Lines 373-377: Names of the bacteria should be italic
Author Response
The manuscript shows interesting and novel results related to the chemical composition and especially physical and functional characteristics of dietary fibres form winemaking by-products (skins, stems, and lees). The study was well designated with the uses of reliable analytical methods. The results are clearly presented with relatively sufficient discussion and explanation.
There are some minor points should be considered/clarified:
- The number of the replicates of the experiments should be mentioned in the Methods
We thank the reviewer´s comment. The number of the replicates of the experiments has been mentioned in the section 2.1 of Material and Methods.
- Table 1: the sum of components calculated based on “g/100 g dry weight” was higher than 100g. This is not reasonable so double-check or explanation for this result must be done. As mentioned in lines 240-243, the dietary fibre content in this study is surprisingly higher than that reported in previous studies with the similar materials. Is there any problem with operation of experiments or calculation of related results?
- Table 2: The sum of IDF components is more than 100 g. Thus double-check or explanation for this result must be done.
- Table 3: Double-check should be done for the statistical comparison of WRC among the samples
Thank of your observations. The data of both tables 1 and 2 has been newly calculated and checked not finding any relevant errors. Certainly, the sum of all components differs to 100% being it mainly attributable to the variability of each determination. In addition, the case of table 1, the amount of the components is referred to dry weight, discarding the moisture of lyophilizate product. Likewise, in the table 2, IDF components is referred to AIR and not to IDF. This error has been corrected in the Table 2.
With respect to the statistical comparison of WRC in the table 3, it has been checked, not observing significant differences due to the high variability found in the values obtained for the lees batch.
- Section 3.3.2: A comparison of phenolic content and antioxidant capacity of materials used in this study with those of some well-known phenolic-rich and strong antioxidant sources would help reinforce the conclusions of the phenolic levels and antioxidant strength of the materials.
A comparison with other studies has been included in section 3.3.2
- Figure 1: Symbol A is hidden, correct it pls.
Symbol A has been corrected
- Lines 373-377: Names of the bacteria should be italic
Thank of your observations. The names of the bacteria have been corrected.
Round 2
Reviewer 2 Report
Linguistic proofreading required.
Author Response
Linguistic proofreading required.
We thank the reviewer´s comment. The English grammar has been newly revised and corrected throughout the manuscript by Proof-Reading-Services.com.